# Structure and Dynamics of Three *Escherichia coli* NfsB Nitro-Reductase Mutants Selected for Enhanced Activity with the Cancer Prodrug CB1954

**DOI:** 10.3390/ijms24065987

**Published:** 2023-03-22

**Authors:** Martin A. Day, Andrew J. Christofferson, J. L. Ross Anderson, Simon O. Vass, Adam Evans, Peter F. Searle, Scott A. White, Eva I. Hyde

**Affiliations:** 1School of Biosciences, University of Birmingham, Edgbaston, Birmingham B15 2TT, UK; 2Institute for Cancer Studies, University of Birmingham, Edgbaston, Birmingham B15 2TT, UK; 3School of Science, RMIT University, Melbourne, VIC 3000, Australia; 4School of Biochemistry, University of Bristol, Bristol BS8 1TD, UK

**Keywords:** nitroreductase, prodrug, flavoprotein, molecular dynamics, CB1954, NfsB

## Abstract

*Escherichia coli* NfsB has been studied extensively for its potential for cancer gene therapy by reducing the prodrug CB1954 to a cytotoxic derivative. We have previously made several mutants with enhanced activity for the prodrug and characterised their activity in vitro and in vivo. Here, we determine the X-ray structure of our most active triple and double mutants to date, T41Q/N71S/F124T and T41L/N71S. The two mutant proteins have lower redox potentials than wild-type NfsB, and the mutations have lowered activity with NADH so that, in contrast to the wild-type enzyme, the reduction of the enzyme by NADH, rather than the reaction with CB1954, has a slower maximum rate. The structure of the triple mutant shows the interaction between Q41 and T124, explaining the synergy between these two mutations. Based on these structures, we selected mutants with even higher activity. The most active one contains T41Q/N71S/F124T/M127V, in which the additional M127V mutation enlarges a small channel to the active site. Molecular dynamics simulations show that the mutations or reduction of the FMN cofactors of the protein has little effect on its dynamics and that the largest backbone fluctuations occur at residues that flank the active site, contributing towards its broad substrate range.

## 1. Introduction

*Escherichia coli* NfsB has been studied extensively, largely due to its potential for cancer gene therapy. The flavoprotein was initially discovered as mutations in the *nfsB* gene make *E. coli* more resistant to the antibiotic nitrofurazone [1]. The natural substrate of this flavoprotein enzyme is not known; however, it reduces a wide range of nitroaromatic compounds to hydroxylamines and also reduces quinones to quinols [2,3], with a bi-bi substituted (ping-pong) enzyme mechanism [2,4]. One of its substrates is the prodrug CB1954, 5-(aziridin 1-yl)-2, 4-dinitrobenzamide, which is reduced to a cytotoxic compound that causes interstrand DNA crosslinks [3]. Transfection of the *nfsB* gene into tumour cells, using an adenoviral vector, followed by the addition of CB1954 causes cell death in tumour cell lines [5,6] and in a mouse model [7]. Phase 1/2 clinical trials of the *nfsB*/CB1954 combination in patients with prostate or liver cancer showed that the adenovirus and the prodrug were both well-tolerated and gave indications of therapeutic benefit [8,9]; however, the Michaelis constant of the enzyme for the prodrug is much higher than the maximum prodrug concentration in serum, greatly reducing its efficacy [10]. While other prodrugs have been made to be used in combination with NfsB [11,12,13,14,15], and the enzyme has also been used with other nitroaromatics, for example, with metronidazole in cell ablation studies [16,17,18], and in bioremediation of TNT [19,20], our studies have focused on understanding the molecular mechanism of the enzyme and on improving the affinity of the protein for CB1954.

In work towards the latter, we previously made a series of mutants using random mutagenesis at nine selected positions within the active site of the protein [21] and combined the best of these to give double mutants [22]. The most active double mutant was T41L/N71S, where at position 41 the threonine is replaced by leucine, and at position 71 the asparagine is replaced by serine. We also developed a method to select mutants that were most active with CB1954, using the SOS activation of lysis in *E. coli* lambda lysogens to release bacteriophage encoding *nfsB* variants [23]. This was used to select the most active mutants in a library containing three random mutations at five different amino acids within the active site. The most active mutant selected was T41Q/N71S/F124T. While the N71S is the only mutation at position 71 that enhances the activity of the protein for CB1954, the F124T mutation is not the most active mutation at position 124, and the T41Q mutation by itself reduces the activity of the protein for CB1954. The mutations together, therefore, show synergistic effects. 

Following our studies, other groups have made multiple NfsB mutants and examined their activity with CB1954 and other nitroaromatic substrates. Linwu et al. [24] examined the reduction of a nitrobenzodiazepine, making several single mutants in active site residues, as well as a double mutant N71S/F124W, with higher activity for the drug. In the structure of the double mutant, the W124 side chain and the F70 side chains rotated, resulting in a larger space above the FMN cofactor. Bai et al. examined single and double mutants at T41, N71, F70 and F124 as well as combinations containing F123A, with CB1945 [25] and other nitroaromatics [26]. They determined the structure of two triple mutants T41L/N71S/F124W (PDB 3X21), which had similar structural effects to the double N71S/F124W mutant studied by Linwu et al. [24], and F123A/N71S/F124W (PDB 3X22) [25], where the smaller F123A mutation gave a wider channel into the binding pocket. Williams et al. made a number of mutants in *E. coli* NfsB, combining the substitutions F70A, F108Y, T41L and N71S, examining their effect on reduction of CB1954 and another similar compound with potential for PET imaging, SN33623, [27]. Of these, F70A/F108Y was the most active, but the introduction of the T41L mutation reduced the activity, and the quadruple mutant T41L/N71S/F70A/F108Y was even less active. The F70A/F108Y double mutation was introduced into the homologous *Vibrio vulnificus* NfsB [18], and the structure of the wild-type enzyme (6CZP) and of this mutant have been determined recently (7UWT). F108 is in a loop between 2 helices and may affect their packing. 

In a previous study, we examined the kinetics in vitro of our most active double (T41L/N71S) and triple mutant (T41Q/N71S/F124T) [28]. These mutants have one hundred-fold and fifty-fold higher specificity constants for CB1954 than the wild-type enzyme, respectively, so they should be much more active at the low concentrations of the substrate possible in vivo. In contrast, the activity of all three enzymes for nitrofurazone is similar, so while the wild-type enzyme shows a twenty-fold preference for nitrofurazone over CB1954, the double and triple mutants show a two-fold preference for CB1954 [28].

In this study, we have determined the structures of these highly active NfsB mutants using X-ray crystallography and measured their redox potentials and kinetics of reduction with NADH. We have further mutated the enzyme to obtain a mutant with higher potency for CB1954 and characterised this in vitro. The structural data on these mutants have been extended with molecular dynamics (MD) simulations of the free oxidised and reduced enzymes to examine the molecular basis for their enhanced activity CB1954. We show that the two mutant proteins have lower redox potentials than wild-type NfsB, which may make the reduction of the nitroaromatic more favourable. In contrast to the wild-type enzyme, where NADH reduction of the bound FMN cofactor is very fast, for the two mutants, the maximum rate of this step is slower than that of reduction of CB1954 by the reduced cofactor FMNH_2_. The structure of the triple mutant shows an interaction between Q41 and T124, explaining the synergy between these two mutations. Our most active mutant in *E. coli* is T41Q/N71S/F124T/M127V, in which the additional M127V mutation enlarges the small channel to the active site. MD simulations show that the mutations or reduction of the FMN cofactors of the protein has little effect on its dynamics, so the enhancements are largely due to small changes in the orientations of the side chains and surrounding residues. 

## 2. Results

### 2.1. Structures of the T41L/N71S NfsB and T41Q/N71S/F124T NfsB Mutants 

The structure of wild-type *E. coli* NfsB, both free [29] and complexed with a number of ligands [4,30,31], has been determined previously, together with structures of the highly similar *Enterobacter cloacae* enzyme [32,33], FRAse from *Vibrio fisherii* [34] and other homologues. Structures of some single mutants of NfsB, namely T41L, N71S, F124K, F124N and F124W, two double mutants N71S/F124K and N71S/F124W [21,24], and two triple mutants T41L/N71S/F124W [26] and N71S/F123A/F124W [25], have also been determined previously, as has the structure of the wild-type and F70A/F107Y double mutant of the *Vibrio vulnificus* NfsB. These structures are largely identical to that of the wild-type protein, apart from at the positions of mutations. 

The enzyme is a homodimer of two 24 kDa subunits with an extensive dimer interface. Each subunit contains five β strands and ten α helices. The two active sites are on opposite sides of the long central helix at the dimer interface, and the FMN is bound to residues from both monomers (Figure 1a). The *re* face of the FMN is solvent exposed at the bottom of a channel, lined by a helix and three loops. While the active site is highly positively charged, the surface of the protein is mainly negatively charged or neutral (Figure 1b). In the structure of the wild-type protein bound to nicotinic acid, T41 NH hydrogen bonds to the acid group of the ligand, N71 forms two hydrogen bonds to the FMN cofactor, and F124 stacks with the aromatic ring of the ligand (Figure 1c).

In this study, structures of the double and triple mutant were both determined with and without bound nicotinic acid (Figure 1d,e, Appendix A). The crystal structures all contain a heterodimer in the asymmetric unit, but, as with the wild-type enzyme, there is little difference between the two subunits, nor between the backbones of the structures with and without nicotinate. For wild-type NfsB, the RMSDs of Cα atoms between the two subunits are 0.39 Å and are 0.3 Å between structures with and without nicotinate [30]. The structure of the T41L/N71S mutant with nicotinate was less well resolved than the other three structures (2.2 Å cf 1.6–1.7 Å), probably due to some mosaicity of the crystal; however, all the structures were well enough resolved to see the mutations and the active sites. 

The double T41L/N71S mutant combines the effects of two single mutations determined previously [21]. On replacing N71 with S71, in most of the structures of all the mutants, a water-mediated hydrogen bond to the FMN cofactor is found from the Oγ of S71 instead of the direct hydrogen bond from the NH_2_ of N71. This water molecule also forms a hydrogen bond with K74, as does N71 NH_2_ in the wild-type enzyme. However, unlike N71, the S71 does not form a hydrogen bond with FMN O4. The leucine side chain at position 41 is larger than the native threonine. It forms a similar hydrogen bond from the backbone nitrogen to the nicotinate oxygen as threonine, but, in addition, it also contacts the nicotinate via one of its methyl groups. The longer side chain of the leucine causes slight shifts in the positions of F124 and W138, each of which is in van der Waals contact with one of the methyl groups of L41 (Figure 1d). 

In the triple mutant, the N71S mutation is the same as in the double mutant; position 41, however, is now a Gln, which is larger than the wild-type threonine and more polar than the leucine of the double mutant, while F124 has been replaced by threonine. In the structures of the free and nicotinate-bound triple mutant, the Q41 side chain forms a hydrogen bond to the T124 hydroxyl group, pulling the residue slightly closer than in the wild-type protein. In the structure with bound nicotinate, the Q41 side chain interacts with the nicotinate ring and is in a different orientation to the side chain of the L41 mutant (Figure 1e). In the structure without nicotinate, there is an acetate group in the active site, and Q41 forms a hydrogen bond to the carboxyl group of acetate so that this ligand is in a slightly different orientation than in the structure of the wild-type protein bound to acetate, determined previously [4], where the hydrogen bond cannot occur.

### 2.2. Redox Potentials of Wild Type, T41L/N71S and T41Q/ N71S/F124T NfsB

To examine whether the mutations affect the redox properties of the protein, we measured the redox potentials of the wild-type enzyme, the double T41L/ N71S mutant, and the triple T41Q /N71S /F124T mutant by monitoring the absorbance changes of the cofactor as the enzyme was reduced and re-oxidised. The absorbance at different wavelengths was fitted either to two single electron reductions (Equation (1)) or to a concerted two-electron reduction (Equation (2)). Figure 2a shows the data and the fits to the two equations for wild-type NfsB, while Figure 2b shows the data and fits to two single electron transfer steps for wild-type and mutant proteins. The fit of the data to two single electron transfer steps in each case is better than to a concerted two-electron reduction; this is particularly evident for the double mutant, for which the data did not fit a concerted reduction (Appendix A). 

The midpoint potential for wild-type NfsB (−217.5 mV) is similar to that measured for the *E. cloacae* nitroreductase enzyme (−210 mV at pH 7.5), but the latter was fitted to a simultaneous two-electron reduction [35]. The double mutant and triple mutants have similar midpoint potentials to each other (−230 mV), slightly more negative than that of the wild-type enzyme and closer to that of free FMN at this pH (−227 mV at pH 7.44 and 20 °C) [36], However, while in the wild type and triple mutant, the redox potential of the oxidised/semiquinone couple (*E’*_1_) is more negative than that of semiquinone/hydroquinone couple (*E’*_2_), the converse is true for the double mutant. This means that in the latter, there is a greater proportion of the semiquinone at equilibrium. However, the spectrum shows no absorbance above 520 nm, expected for a neutral semiquinone species, nor any indication of a semiquinone in the visible spectrum of any of the proteins (Appendix A).

### 2.3. Stopped-Flow Kinetics

The crystal structure of the protein [30] and previous kinetics studies [21,37] show that *E. coli* NfsB and the related *E. cloacae* enzyme [38] have a bi-bi substituted enzyme mechanism, where the NAD(P)H first reduces the FMN cofactor of the protein, and then the nitroaromatic or quinone binds and is reduced in turn. In previous work, we measured the oxidative half-reaction of the double and triple mutants with CB1954 and nitrofurazone [28]. The stopped-flow kinetic analysis of the oxidative half-reaction with CB1954 confirmed that the mutants had a much faster second-order rate constant, k/K_d_, than the wild-type enzyme, consistent with the steady-state kinetics; however, the full kinetic curves could not be obtained due to the limited solubility of the substrates. Hence, we were not able to obtain values for k and K_d_ separately. The second-order rate constants for nitrofurazone are much more similar for all of the enzymes [28]. 

We here show the reductive half-reaction of the mutants and wild-type enzyme with NADH using stopped-flow kinetics (Figure 3, Appendix A). In the reductive half-reactions, the full hyperbolic curves were obtained, allowing the determination of the dissociation constant of NADH, K_d_, and the maximum rate, k, for all three enzymes. The wild-type enzyme shows about a five-fold faster rate of reduction with NADH than either of the mutants. In contrast, the dissociation constant for NADH is about double that for the triple mutant and slightly higher than that for the double mutant. This leads to a two to three-fold higher second-order rate constant k/K_d_ for this reaction with the wild-type enzyme over the triple mutants and double mutant, respectively.

### 2.4. Quadruple Mutants

To try to further improve the activity with CB1954, mutations were made at more positions around the active site of NfsB. The positions M127, H128, and W138 were mutated to all possible amino acids (Figure 4), with F, N, K, or T at F124 and wild-type or preferred combinations at T41, S40, F70, and N71. Mutants with improved activity for CB1954 were selected by introducing the mutants into lambda phage, forming *E. coli* lysogens, and using the NfsB reduction of CB1954 to activate the SOS response, thus releasing phages enriched with more active enzymes [23]. 

After 15 rounds of selection, the mutations contained L or Q at position 41, only S at 71, F or T at 124, and a variety of hydrophobic residues at 127. IC_50_ colony assays suggested that the most active mutants were T41Q/N71S/F124T with V, T, I, or Y at position 127 (Appendix A). Figure 5 shows the CB1954 sensitivity of *E. coli* lysogens carrying *nfsB* with different mutations, measured by counting the number of colonies formed on replica plating the lysogens at a series of concentrations of the prodrug. With wild-type *nfsB*, only 50% of the original number of colonies grew at about 210 µM CB1954 compared to the number in the absence of the prodrug. The mutations that are more reactive with the prodrug make the lysogens more susceptible to killing by CB1954. *E. coli* lysogens expressing the NfsB double and triple mutants are killed at much lower CB1954 concentrations, with the triple mutant being more active than the double mutant. The quadruple mutation containing M127I shows higher sensitivity to CB1954 than the triple mutation, while the T41Q/N71S/F124T/M127V mutant is the most sensitive, about twice as sensitive as the triple mutant and fifty times as sensitive as wild type NfsB. On repeating the assay on different days, the absolute values for the IC_50_s varied slightly, but the relative efficacies of the mutants remained constant.

The T41Q/N71S/F124T/M127V mutant was purified, and its steady-state kinetics for the reduction of CB1954 were measured at different NADH and CB1954 concentrations, as shown in Figure 6. The data were fitted to the kinetic equation for a bi-bi substituted enzyme (Equation (4)) to give the Michaelis parameters (Appendix A). The mutant has a higher k_cat_ than previous mutants, while its K_m_ for CB1954 is much lower than for wild-type NfsB but between that of the double and triple mutants [28]. The specificity constant for CB1954, k_cat_/K_m_, that determines the rates at low substrate concentration is 0.57, eighty-fold higher than the wild-type but not quite as high as for the double mutant. In contrast, the K_m_ for NADH is lower than that of these mutants, leading to a higher specificity constant for NADH, but the K_m_ for NADH of the wild-type protein is still lower. 

The structure of this mutant was determined at 2 Å resolution (Figure 7c, Appendix A). The side chain change of this additional mutation is too far away to have a direct effect on the active site of NfsB; however, the M127V mutation enlarges a small channel leading to the active site. In NfsB, the main opening to the active site is as shown in Figure 1; however, there is a smaller opening to the site, approximately at right angles to the first one, shown in Figure 7. In the wild-type protein, a protein bridge, formed by F124 and Y68 side chains, is above this small channel so that it is only large enough for solvent and small ions (Figure 7a). In the triple mutant, two smaller bridges form between M127 and Y68 and between N67 and F123 (Figure 7b). However, the mutation of M127 to the smaller valine allows the side chains of Y68 and N67 to move slightly, opening the channel (Figure 7c). This could allow larger substrates to be accommodated in the active site. 

### 2.5. Molecular Dynamics (MD) Simulations

MD simulations were carried out to examine the fluctuations of the Cα atoms of the residues in the mutant and wild-type proteins and the distances between selected residues in the minor channel. In each case, three simulations were run over 200 ns based on the structures above, removing any ligands in the active site. As the protein is a homodimer with active sites between the subunits, this gave three results for each subunit, arbitrarily called subunit one and subunit two, and three results for each active site, arbitrarily labelled site 1 and site 2. In previous MD simulations, we had shown that the orientation of bound ligands in *Enterobacter cloacae* NfsB [39] and in *E. coli* NfsA [40] changes when the FMN cofactor is reduced. We, therefore, ran simulations with the reduced cofactor, FMNH^−^, in the active sites as well as with the oxidized cofactor, FMN, in the sites. 

All the structures remained stable over the dynamics runs. The root-mean-square deviations (RMSDs) of the Cα atoms in the entire structures show an initial rise for about 20 ns and then a stable difference of 1.3 ± 0.1 Å from the initial structures. Figure 8 shows the root-mean-square fluctuations (RMSFs) of the individual Cα atoms averaged over the three runs in each site. In all cases, Cα atoms from residues 65–70, 108–134, and 196–204 show higher RMSFs than others, with residues 65–67, 109–112, and 131–133 showing fluctuations greater than 1.5 Å and maximum RMSFs at residues 66, 109, and 131–132. A slight difference between the averages in site 1 and site 2, within each plot, is within the standard deviations between the runs. There are no significant differences between the four enzymes and between the oxidised and reduced enzymes, showing that the mutations or reduction of the protein has little effect on the dynamics of the enzyme. 

To examine the fluctuations in the size of the minor channel, distances between the centre of mass of the side chains in selected residues were measured over the course of the simulations (Appendix A). For Phe and Tyr, this was from the centre of mass of the ring carbons; for Asn, it was between CG, OD1, and ND2. For Met, CG, SD, and CE were used, while for Val, it was the centre of mass between CG1 and CG2. The distances between F124 and Y68 over the MD simulations were extracted in both active sites for wild type and T41L/N71S. The distances between F123-N67 and M/V127-Y68 were extracted for all the mutants in all the dynamics runs. For wild type and T41L/N71S the distances, on average over the six sites, are a little longer in the reduced enzymes than the oxidised enzymes, but the difference is within the standard deviations within each dynamic run. This difference between oxidised and reduced enzymes was not seen for the other mutants. For the M127V, the average distances between the centre of mass of V127 and Y68 were longer than the distance from the centre of mass of M127 and Y68 for the other proteins, as V has a shorter side chain than M; however, even here, the differences in the distances in the mutants is within the standard deviations within each dynamic run. 

## 3. Discussion

In this paper, we have examined the structures of three *E. coli* NfsB mutants. At position 71, all the mutants contain S71 instead of N71. The effect of this is identical to that seen in all the structures with this mutation examined previously [21,24,25] in that a direct hydrogen bond from the N71 side chain to the FMN cofactor is replaced by a water-mediated hydrogen bond from S71 (Figure 1). This change in the bonding to the FMN is likely to be the cause of the difference in the redox potentials of the double and triple mutants from that of the wild-type protein measured here. Both mutants were found to have the same mid-point potential, slightly more negative than that of the wild-type protein. While the mid-point potential of the wild-type enzyme is similar to that measured for the *E. cloacae* nitroreductase enzyme [35], the reduction of the latter was shown to proceed by a concerted 2-electron reduction, in contrast to our measurements, which better fit two single electron transfer steps, particularly in the case of the T41L/N71S mutant (Figure 2). This is in line with previous computational studies that suggested the electron transfer from reduced FMN to CB954 should occur by two sequential single electron reductions [41,42]. Despite this, there is no evidence of a semiquinone species in our redox spectra, and it is unclear why the measurements for the *E. cloacae* and *E. coli* nitroreductase enzymes differ. 

The lowering of the redox potential would help electron transfer to prodrugs but make the reduction by NADH less thermodynamically favourable. Our stopped-flow results (Figure 3) show that the wild-type enzyme has a much faster rate of reduction by NADH than the two mutants. This is in contrast to our previous stopped-flow assays of these enzymes with CB1954 [28], where the mutants showed much higher rates of reaction than the wild type. In the latter experiments, we were not able to obtain the full kinetic curves due to the high dissociation constants for the nitroaromatics and their poor solubility, but we did obtain apparent first-order rate constants over 400 s^−1^ for the mutants at high CB1954 concentrations. This is higher than the maximum rates for NADH reduction of the mutants (Figure 3, Appendix A). In the structures of the proteins bound to nicotinate, both L41 and Q41 interact with the nicotinamide ring and are likely to also interact with CB1954. The stopped-flow data suggests that while the mutations have improved the binding orientation of the prodrug, they may have compromised that of the NADH cofactor so that it is no longer optimal for direct hydride transfer to the N5 of the FMN. However, while at high concentrations of CB1954, the NADH step is limiting for the mutants, the in vivo concentrations of NADH are high compared to the achievable concentrations of CB1954 so that in vivo, the NADH half of the reaction remains the faster step. The improved rates of reduction of CB1954 by the mutations make them faster than the wild-type enzyme at these low CB1954 concentrations. 

At T41, the T41L mutation in the double mutant shows the same effects as seen in the single T41L mutant [21] and the T41L/N71S/F124W mutant studied by Bai et al. [25] (Figure 1c). In the triple and quadruple mutants, T41Q and F124T were selected. The T41Q single mutation is detrimental to CB1954 activity; while F124T is not the most active single substitution at F124, twelve other substitutions were found to be more active [43]. The three mutations in the triple mutant act synergistically [23]. The crystal structure of the T41Q/N71S/F124T mutant (Figure 1d) shows a hydrogen bond between Q41 and T124, explaining the co-operativity between these two residues. 

Despite making mutations at eight sites to multiple other amino acids, the most active quadruple mutants isolated contained the same three mutations as in the triple mutant, with an additional mutation at M127. The most active mutant was T41Q/N71S/F124T/M127V, in which a small channel into the active site is enlarged compared to the other mutations, which may allow access by small substrates as well as ions. This is the same channel that was found to be widened in the F123A/N71S/F124W mutant of Bai et al. [25], but it is widened closer to the FMN, deeper into the channel, between 124-Y68, in the T41Q/N71S/F124T/M127V mutant (7.9 Å cf 5.2 Å) as the T124 replaces the larger W124, but less wide closer to the outside, at residues 123-N67 (6.4 Å cf 11.4 Å). In the structure of the triple mutant T41Q/N71S/F124T, the side chain of M127 is in the channel, and the side chain of N67 is rotated towards it, so the channel is narrower than in the quadruple mutant with M127V; however, both N67 and M127V show large RMSFs in the MD calculations (Figure 8) and are likely to move within the channel. 

The steady-state kinetics of reduction of CB1954 with the T41Q/N71S/F124T/M127V mutant shows the highest k_cat_ of all the mutants to date and a much lower K_m_ for CB1954 than the wild-type enzyme (Appendix A), so it should be much more active with the prodrug. Its K_m_ for NADH is lower than for the other mutants but higher than that of the wild-type enzyme. The specificity constant k_cat_/K_m_ for CB1954, which determines the rate at low CB1954 concentrations for this mutant, is higher than that for the triple mutant and slightly lower than that of the double mutant, which, however, has the lowest k_cat_/K_m_ for NADH. The quadruple mutant has the highest activity in the colony-forming assays in *E. coli*, with about a fifty-fold increase in sensitivity to CB1954 over the wild-type enzyme, similar in magnitude to its decrease in K_m_ for CB1954 (Figure 5, Appendix A). This suggests it could be a promising prodrug-activating enzyme in combination with CB1954; however, further assays are required to determine its efficacy in human cells. 

All the structures have very similar backbone conformations, which show minimal changes during the MD simulations. However, some of the Cα atoms have large RMSFs (Figure 8). Residues 67–70 are very dynamic in all the structures, and the Phe 70 and Phe 124 side chains, which were proposed to gate the access to the active site, have been found in different orientations in different crystal structures of the wild-type protein [30]. Residues 66–68 are on a loop at the mouth of the channel, while 69–70 are on the first turn of a helix, with 71 binding the FMN. Residues 108–134 also have high RMSFs in the MD calculations. These residues, which include F124 and M127, are in a loop and a long helix lining the active site, in an insertion in the NfsB family proteins that is not found in other families within the nitroreductase superfamily of proteins [44]. In this region, residues 131–132 have the highest RMSFs and are at the mouth of the channel, opposite residue 67. Residues 109–112, which are also highly dynamic, are at the other end of this helix and close to the FMN side chain. Residues 109 and 110 are in a loop, while 112 is in the first turn of a helix. In the highly active F70A/F108Y double NfsB mutant of Sharrock et al. [45], the mutations are likely to affect the dynamics of the residues across the active site of the protein. The mutations may also affect the orientation of the neighbouring residues 107 and 71. In the structure of *Enterobacter cloacae* NR bound to NAAD (nicotinic acid adenosine dinucleotide) (5J8D), the backbone of 107 interacts with the amine of the adenine cofactor, while the side chain of N71 is close to the ribose of the nicotinic acid [33]. It is likely that the F70A and F108Y mutation affect the binding or orientation of the NAD(P)H within the binding site, as found with the mutants in this study, so that reduction of NAD(P)H could become rate limiting at lower NAD(P)H concentrations if these mutations are combined. Hence, additional T41L/N71S mutations make the F70A/F108Y enzyme less active [27]. Residues 196–204, which are also dynamic, are in a loop near the C-terminus of the protein that crosses the dimer interface, at the bottom of the active site, with residues in van der Waals contact to the adenosine half of NAAD bound to *Enterobacter cloacae* NR [33] and the neighbouring residues K205 and R207 interacting with the phosphate group of FMN. The dynamics of the residues lining the active site may allow the broad range of substrates to be accommodated. 

Despite the lower dissociation constants of the mutants for CB1954 than that of the wild type, we were unable to obtain crystals of the proteins with bound CB1954 or related dintrobenzamide prodrugs, even at neutral pH where the prodrugs are stable. Given that the binding of nitroaromatic antibiotics is reversed in oxidized and reduced NfsB [30] and NfsA [40], it is possible that the prodrugs only bind to the reduced enzyme or with NAD(P)^+^ in one of the two active sites [39,46]. Future work will concentrate on modelling the interactions with the reduced enzymes. 

## 4. Materials and Methods

### 4.1. Protein Expression and Purification

Wild-type NfsB [30] and the T41L/N71S [22,28] and T41Q/N71S/F124T [23] NfsB mutants were made as described previously. They were recloned into pET11c vectors and expressed as the native proteins in *E. coli* BL21 (λDE3) cells. The T41Q/N71S/F124T/M127V was initially purified from a UT5600 *E. coli* lysogen, described below, but later also recloned into pET11c. The proteins were purified using chromatography on Phenyl Sepharose and Q-Sepharose columns as described previously [3,30] and were >90% pure based on Coomassie-stained PAGE. Protein concentrations were determined by Bradford assays, calibrated against bovine serum albumin [47], or by determining the absorbance at 280 nm, where both the protein and the cofactor absorb, and correcting for excess FMN by measuring the absorbance at 454 nm, where only FMN absorbs. The molar absorbances used were 12, 200 M^−1^ cm^−1^ for FMN at 454 nm, 20,970 M^−1^ cm^−1^ for FMN at 280 nm and 22, 460 M^−1^ cm^−1^ for NfsB at 280 nm, based on its amino acid composition [48]. To check that the purified proteins were saturated with FMN, the ratio of absorbance at 280 nm to that at 454 nm was measured.

### 4.2. Protein Crystallisation

NfsB mutants were crystallized using conditions similar to those used previously [4,21,30] with 10–18% PEG 4000, 50–200 mM sodium acetate buffer pH 4.6, 15 mM nicotinic acid (where present), and 15% ethylene glycol. X-ray diffraction data were collected at synchrotrons of the European Synchrotron Radiation Facility (ESRF), Grenoble, France. Points were indexed, integrated and processed using iMOSFLM [49] or XDS [50]. Datasets were combined and scaled using pointless and scala [51], and the quality of the data was assessed using xtriage [52]. All structures were solved by molecular replacement with Phaser [53], using the starting model PDB ID 1ICR for wild-type NfsB [30], replacing the mutated residues with alanines so the density for the mutation sites could be judged before modelling. Structures were refined using Refmac5 [54,55] or PHENIX [52], in the CCP4 suite of programmes [56,57,58], and in PDB_Redo [59]. Model building and modification were performed using Coot [60]. Final models were validated using Molprobity [61]. 

Structural figures were drawn with Chimera 1.16 [62], and the electrostatic surfaces were calculated using APBS [63].

### 4.3. Redox Potentiometry

Potentiometric titrations were performed using a modified quartz EPR OTTLE cell, equipped with platinum working and counter electrodes and an Ag/AgCl reference electrode (BASi, West Lafayette, Indiana, USA), in concert with a Biologic SP-150 potentiostat (Bio-logic, Seyssinet-Pariset, France) and Cary 60 UV/visible spectrometer (Agilent, Santa Clara, CA, USA). Titrations were performed in OTTLE buffer (50 mM phosphate, 500 mM KCl, 10% glycerol, pH 7.5) using a protein concentration of 50–150 μM. The redox mediators usually used in titrations are quinones that were found to react with the enzyme, so the experiments were carried out in the absence of redox mediators. The potential was initially changed from −50 mV to −350 mV in steps of 20 mV and full UV spectra taken after equilibration at each step to observe the reduction of the protein. The titration was then reversed, changing the potential to −335 mV and then from −320 mV to −90 mV, in steps of 20 mV and finally to −60 mV. Each titration was carried out in duplicate. 

The reduction potentials were determined by fitting the absorbance at 450 nm as a function of potential either to two single 1-electron Nernst equations (Equation (1)) or to a concerted 2-electron Nernst equation (Equation (2)) using Sigmaplot14, assuming the temperature is 25 °C. The slow reduction in the absence of mediators (waiting 1 h between each step) led to some differences between the reduction and the oxidation titrations, so first, each titration was fitted separately to determine the absorbance of the oxidised and reduced solution. The absorbance of each point was then scaled to give the proportion of protein oxidised, and the data from both half titrations for the two replicas were fitted to the equations with *a*, *b* and *c* fixed to 1, 0.5 and 0, respectively. Reported reduction potentials are quoted vs. the Nernst hydrogen electrode (NHE).
(1)$$A=\frac{a\times 10^{\frac{\left(E-{E^{\prime }}_{1}\right)}{59}}+b+c\times 10^{\frac{\left({E^{\prime }}_{2}-E\right)}{59}}\;}{1+10^{\frac{\left(E-{E^{\prime }}_{1}\right)}{59}}+10^{\frac{\left({E^{\prime }}_{2}-E\right)}{59}}}$$
(2)$$A=\frac{a\times 10^{\frac{\left(E-E_{m}\right)}{29.5}}+c}{1+10^{\frac{\left(E-E_{m}\right)}{29.5}}}$$

*A* is the absorbance observed at the potential *E*; *E′*_1_ and *E′*_2_ are the midpoint potentials for the oxidized/semiquinone and semiquinone/reduced couples, and *E_m_* is the midpoint potential for a concerted two-electron transfer. The constants *a*, *b*, and *c* are the absorbances of the quinone, semiquinone and hydroquinone, respectively.

### 4.4. Stopped-Flow Kinetics 

Stopped-flow kinetic experiments were performed with an Applied Photophysics DX17MV spectrophotometer, with a dead time of 1.0 ms, monitoring the reduction of the enzyme-bound FMN at 454 nm. The optical path length used was 2 mm, with 4 mm monochromator slits. The enzymes, at an initial concentration of 10 µM in 10 mM Tris HCl buffer, pH 7.0, and the NADH (at concentrations of 25–1000 µM) were placed in separate syringes with a stop volume of 120 µL. At each NADH concentration, 4–8 duplicate reactions were performed. The absorbance change of each reaction as a function of time was fitted to a single exponential, giving a pseudo-first-order rate constant, k_1_. These rate constants were then plotted against substrate concentration and fitted to the equation of a hyperbola to determine the maximum rate of reduction of the enzyme k and the dissociation constant of the NADH, K_d_ NADH.

### 4.5. Generation and Selection of Further Mutants

A lambda phage library was generated containing a variety of *nfsB* genes under the control of a p*tac* promoter in λJG16C2 [43], containing a kanamycin resistance cassette. The library contained random base substitutions at the codons for M127, H128, and W138, with codons encoding substitutions at S40, T41, F70, N71 and F124, which were previously found to give higher activity against CB1954. These were S or A at S40, T, L, Q, or G at T41, F or A at F70, N or S at N71, and F, N, K, or T at F124. The library was generated by PCR joining of fragments and mixtures of primers containing NNN at the codons for mutagenesis. The backbone templates contained either wild type, N71S, T41-N/L/Q/G, N71S, S40A, S40A/F70A, and F70A *nfsB* genes. The final full-length PCR products were digested with *Sfi*I and ligated into the vector. The total library complexity was ~1.47 × 10^6,^ and the titre of the combined libraries was ~1 × 10^8^. The library was screened by PCR for insertion of the *nfsB* gene, and a sample of clones was sequenced. This contained mutations at each expected position. To preserve the diversity of the library, it was first amplified by a round of lytic growth to give ~1 × 10^9^ plaque-forming units/mL.

Quadruple mutants were selected as described previously [23]. The phage library was used to infect the *recA*^+^ *nfsB*^−^ strain of *E. coli*, UT5600, to generate lysogens, which were plated on LB agar containing 30 µg/mL kanamycin. Cultures from these were grown at 37 °C to an OD_600_ of 0.1 in LB with kanamycin, when 0.1 mM IPTG was added to induce expression of NfsB. At OD_600_ 0.5, 1 mL of culture was diluted 5-fold in LB/kanamycin, 0.1 mM IPTG and 0–100 µM CB1945, for 15 min. The reduction of CB1954 catalysed by NfsB induces the SOS response, causing lysis and phage release. After the transient exposure to CB1945, 10 µL of the culture was added to 10 mL fresh LB and incubated for a further 1 h to allow the lysogens to complete the lytic cycle. The released phage is enriched for more active NfsB, and the phage was harvested and used to infect fresh UT5600 cells for the next round of selection, decreasing the concentration of prodrug from 100 µM to 50 µM and then 25 µM CB1954 for increasing stringency at later rounds of selection. 

### 4.6. Analysis of Selected Phages 

Plaques of selected phages were picked for analysis. Primers binding to the lambda genome flanking the *nfsB* insert were used to amplify the *nfsB* DNA for each plaque, and the PCR fragment was sequenced. Representative phages containing each *nfsB* variant were used to generate lysogens and tested for CB1954 sensitivity by replica plating from a liquid culture in 96-well plates onto LB agar containing 30 µg/mL kanamycin, 0.1 mM IPTG, 50 mM Tris HCl pH 7.5, and a range of CB1954 concentrations. The more sensitive lysogens were analysed further in a colony-forming assay.

Lysogens were grown in LB/kanamycin; at an OD_600_ of 0.1, IPTG was added to a final concentration of 0.1 mM to induce *nfsB* expression. At OD_600_ 0.5, the culture was diluted 250,000-fold, and 100 µL was spread onto LB agar plates containing 50 mM Tris HCl pH 7.5, 0.1 mM IPTG, 30 µg/mL kanamycin, and a range of CB1954 concentrations. Colonies were counted after overnight incubation at 37 °C. The number of colonies at each prodrug concentration was fitted to a sigmoidal curve in Sigmaplot 14 (Equation (3)) to obtain the scaling constant, a, and *x*_0_—the concentration of prodrug at which 50% of the colonies survive, here defined as the IC_50_.
(3)$$y=\frac{a}{1+e^{\frac{-\left(x-x_{0}\right)}{b}}}$$

### 4.7. Steady-State Kinetic Enzyme Assays 

Steady-state kinetic assays with CB1954 were monitored spectrophotometrically at 420 nm, over 1–2 min, as described previously [4], using a molar absorbance change of 1200 M^−1^ cm^−1^. CB1954 was dissolved in a 2:7 mixture of NMP:PEG 300; kinetic experiments included final concentrations of 1.1% NMP and 3.9% PEG 300 in 10 mM Tris HCl pH 7.0. Reactions were carried out at 25 °C, initiated by the addition of a small quantity of enzyme (~10 nM). 

For each reaction, the initial rate (*v*) was calculated for a range of concentrations of one substrate (A) whilst keeping the concentration of the other substrate (B) constant. Kinetic measurements were collected over a range of concentrations of both substrates and fitted to Equation (4), describing the overall kinetics for a bi-bi substituted enzyme (ping-pong) reaction, by nonlinear regression with equal weighting of all points, using the programme Sigmaplot 14 (Systat software, San Jose, CA, USA). [E] is the enzyme concentration.
(4)$$\frac{v}{\left[E\right]}=\frac{k_{\mathrm{cat}}\left[A\right]\left[B\right]}{K_{\mathrm{mA}}\left[B\right]+K_{\mathrm{mB}}\left[A\right]+\left[A\right]\left[B\right]}$$

### 4.8. Molecular Dynamics (MD) Simulations

The initial structure for wild-type NfsB was taken from the 1YLR X-ray crystal structure [4]. Initial structures for the mutants came from the crystal structures of the nicotinic acid-bound enzymes in this study. In each case, all ligands were removed. Hydrogen atoms were added using the tleap program of AmberTools 21 [64] according to physiological pH and solvated with a TIP3P water box. Models of oxidized and reduced FMN were taken from previous work [39], and the ff14SB force field [65] was applied to all amino acids. Simulations were run either with both FMN residues oxidized or both FMN reduced. All simulations were performed using the GROMACS 2022 MD software [66]. ACEPYPE [67] was used to convert the topology from AMBER format to GROMACS format. Prior to the MD simulation, a molecular mechanics minimization was performed on each structure, employing the steepest descent method, with a maximum force convergence criterion of 20 kJ mol^−1^ nm^−1^. Each simulation was equilibrated by 1 ns of constant pressure MD at 300 K and 1 bar, with position restraints of 1000 kJ mol^−1^ nm^−2^ applied to the protein alpha carbons. Unrestrained simulations, with coordinates saved every 10 ps, were run for 200 ns, in triplicate, with temperature maintained at 300 K by a Nose–Hoover thermostat and pressure maintained at 1 atm with a Parrinello–Rahman barostat. The LINCS algorithm was applied to all bonds to allow a 2 fs timestep. A 10 Å cutoff was applied to electrostatic and van der Waals interactions, with the particle-mesh Ewald scheme applied to long-range electrostatics. The ptraj tool of AmberTools21 and VMD 1.9.3 [68] were used for analysis, which was carried out on the final 100 ns of each trajectory.

## 5. Conclusions

This paper presents the structures and dynamics of three NfsB mutants with enhanced activity for CB1954. Together with our previous paper [28], it is a comprehensive examination of two of the mutants, T41L/N71S and T41Q/N71S/F124T. In the structure of the latter, a hydrogen bond between Q41 and T124 explains the synergy between the mutation T41Q, which is otherwise detrimental, and F124T, which is less beneficial than other mutations at F124. This synergy is sufficient in all further mutants selected for their greater activity for CB1945, from a combination of possibilities at 8 positions with 4 possible amino acids at both 41 and 124, having these two mutations at these positions. The T41L/N71S and T41Q/N71S/F124T mutants both have a slightly lower redox potential than the wild-type protein. This is most likely due to the N71S mutation, where a direct hydrogen bond to the FMN cofactor is replaced by a water-mediated bond. The stopped-flow kinetic studies show that, in contrast to the wild-type enzyme, for these mutants the maximum rates of reaction with the cofactor NADH are lower than the rates of reduction of CB1954 measured previously [28]. At the low CB1954 concentrations achievable, the overall rates are still limited by the reduction of the prodrug rather than the NADH step, and so in vivo, the two mutants are more active than the wild type. The detrimental effects of mutations on cofactor binding and reaction may explain why some combinations of individually beneficial mutations can be detrimental overall, even when the mutations are distant from each other. 

Following the studies of the double and triple mutants, we selected and characterised a quadruple mutant with even higher activity for CB1954 in *E. coli* cell-killing assays. The mutations, T41Q/N71S/F124T/M127V, enlarge a small second channel to the active site while increasing the k_cat_ of the enzyme and decreasing its K_m_ for CB1954. While there is little change in backbone structure or backbone dynamics of NfsB on mutation or on reduction of the cofactor, the residues that show the most fluctuations are those lining the active site of the enzyme that control substrate and cofactor binding, including this second channel. These fluctuations may allow the binding of a broad range of substrates to this enzyme. 

Given this broad substrate range, it is hoped that the findings of this study will help in the rational design of nitroreductase enzymes, both for cancer therapy and for many other reactions. The effects of mutations on each half of the reaction need to be considered for any protein with a substituted enzyme mechanism, where the cofactor and substrate share the same active site and may limit the maximum efficacy of mutations.

## Figures and Tables

**Figure 1 ijms-24-05987-f001:**
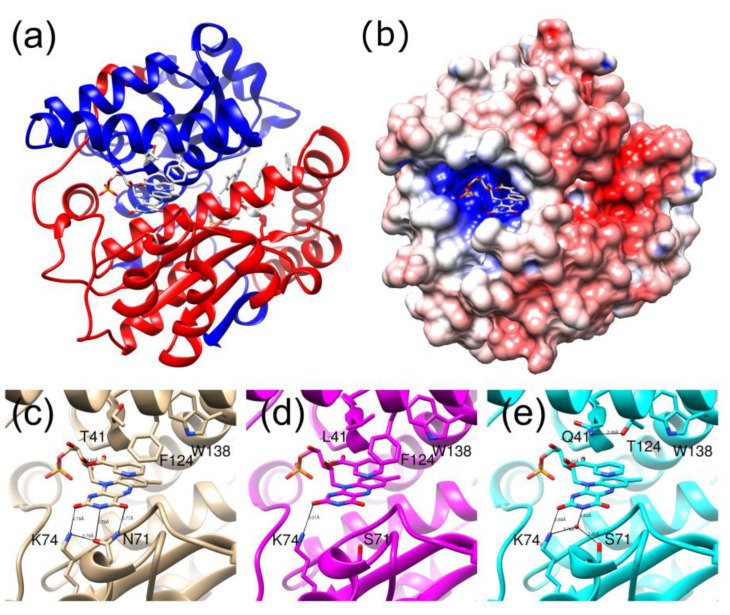
Structure of *E. coli* NfsB wild-type and mutants bound to nicotinic acid. (**a**) Ribbon diagram of wild-type enzyme from 1ICR [30]. One subunit is coloured red, the other blue. The FMN, bound nicotinic acid, and mutated sites are shown in full and in white with CPK colours. (**b**) Electrostatic surface representation of the protein in the same orientation as in (**a**). Negative charges are shown in red and positive charges in blue. (**c**) close up of the active site of the wild-type protein, in the same orientation as in (**a**) with the FMN, nicotinic acid, mutated sites, as well as the neighbouring K74 and W138, shown in full in CPK colours. Hydrogen bonds between residues are shown with black lines. (**d**) The same region as in (**c**) in the T41L/N71S mutant. (**e**) The same region as in (**c**) in the T41Q/N71S/F124T mutant.

**Figure 2 ijms-24-05987-f002:**
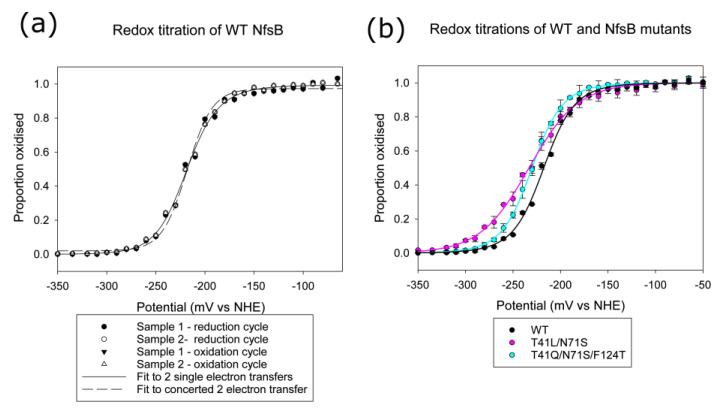
Potential titration of NfsB. Titrations were performed with 50–100 µM protein in 50 mM phosphate buffer, pH 7.5, 500 mM KCl, 10% glycerol, in the absence of redox mediators, with two aliquots of the same enzyme preparation. (**a**) Titration of wild-type NfsB; circles show the reduction cycles, triangles show the oxidation cycle, black symbols show sample 1, and white symbols show sample 2. The solid line shows the fit of the data to 2 single electron transfer steps (Equation (1)) with potentials −231 mV and −204 mV, while the dashed line shows the fit to a concerted 2-electron transfer (Equation (2)) with potential −218 mV. (**b**) Titration of wild-type NfsB (black circles), T41L/N71S NfsB (magenta circles), and T41Q/N71S/F124T NfsB (cyan circles). The circles show the scaled data from both titrations for each protein, with their error bars, while the lines show the fit of the data to 2 single electron transfer steps (Equation (1)). Wild-type enzyme was fitted as in (**a**), T41L/N71S was fitted to potentials −215 mV and −252 mV, and T41Q/N71S/F124T was fitted to −245 mV and −215 mV.

**Figure 3 ijms-24-05987-f003:**
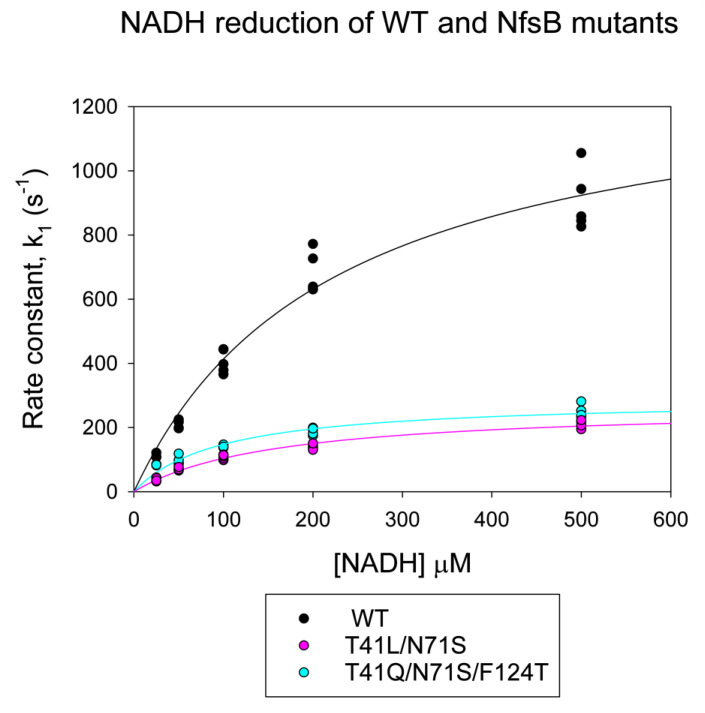
Plots of pseudo-first-order rate constants k_1_ vs. NADH concentration for the reduction of NfsB and mutants by NADH, monitored by stopped-flow. Rate constants were calculated by fitting the absorbance at 340 nm with time after addition of NfsB to an exponential equation. The reactions were done in a 10 mM Tris HCl pH 7 buffer at 25 °C with 5 µM NfsB or mutants. Circles show the data points, while lines show the fits to a hyperbola, with the parameters in Appendix A. Black circles and lines represent wild-type NfsB, magenta circles and lines represent T41L/N71S NfsB, and cyan circles and lines represent T41Q/N71S/F124T NfsB.

**Figure 4 ijms-24-05987-f004:**
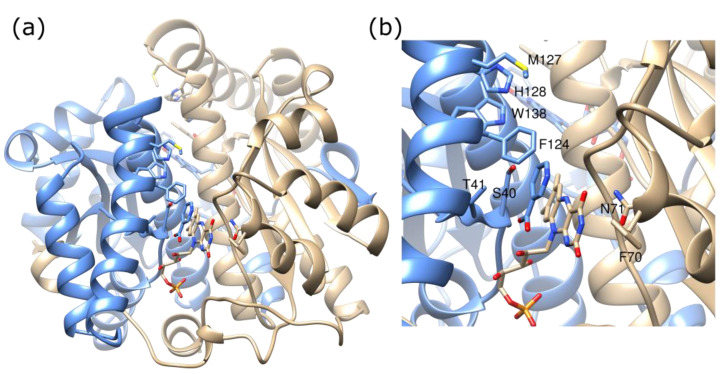
Ribbon diagram of wild-type protein showing positions mutated. (**a**) Structure of full protein with one subunit in blue and the other in tan, from 1ICR [30]. The side chains of mutated residues are shown in full and coloured in CPK colours, as are the FMN and bound nicotinate. (**b**) Close-up of the active site, in the same orientation as in Figure (**a**), with the mutated residues labelled.

**Figure 5 ijms-24-05987-f005:**
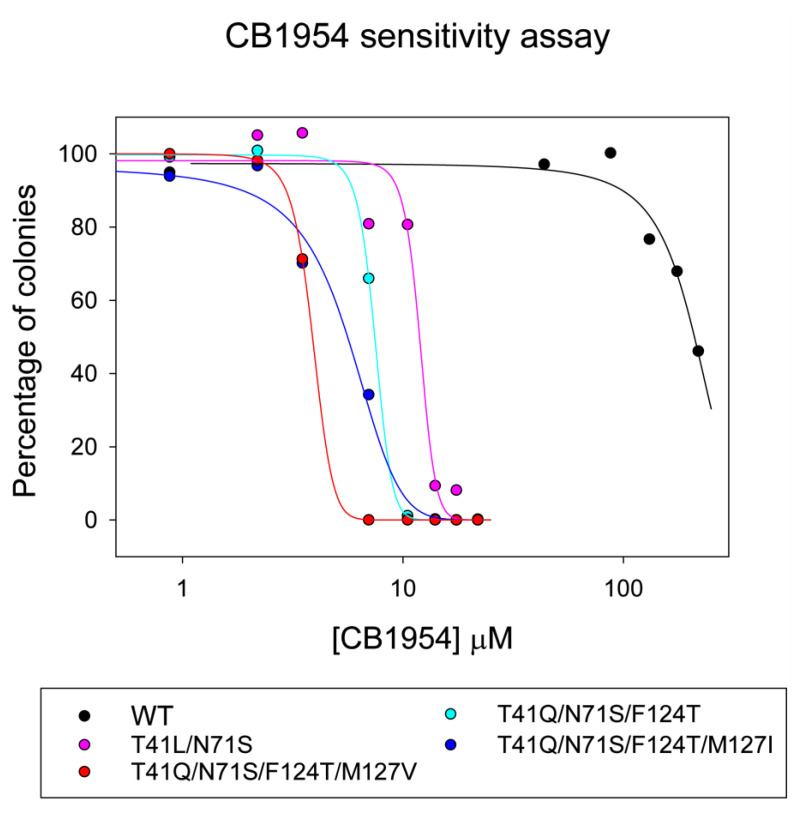
Survival at different concentrations of CB1954 of lysogens expressing either wild-type or mutant NfsB. Black circles—wild type protein, magenta circles—T41L/N71S NfsB, cyan circles—T41Q/N71S/F124T; blue circles—T41Q/N71S/F124T/M127I; red circles—T41Q/N71S/F124T/M127V. Lines show the fits to a sigmoidal curve (Equation (3)).

**Figure 6 ijms-24-05987-f006:**
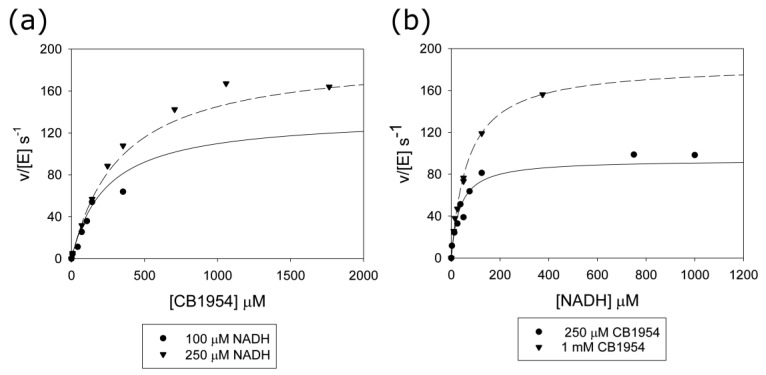
Steady-state rates of the reduction of CB1954 with NADH catalysed by T41Q/N71S/F124T/M127V NfsB. The reactions were done in 10 mM Tris HCl pH 7.0 buffer containing 1.1% NMP and 3.9% PEG 300 at 25 °C and monitored at 420 nm. (**a**) Reactions at constant NADH concentrations, varying CB1954 concentrations. Circles—initial rates of reaction at 100 µM NADH; triangles—250 µM NADH. (**b**) Reactions at constant CB1954 concentrations, varying the NADH concentrations. Circles—initial rates of reaction at 250 µM CB1954, triangles—1 mM CB1954. The lines show the fit of the data to Equation (4), with k_cat_ 270 s^−1^, K_m_ CB1954 460 µM, and K_m_ NADH 100 µM. The dashed lines are the calculated rates for the higher concentration of varied substrate, with the solid lines showing the calculated rates for the lower concentrations of varied substrate.

**Figure 7 ijms-24-05987-f007:**
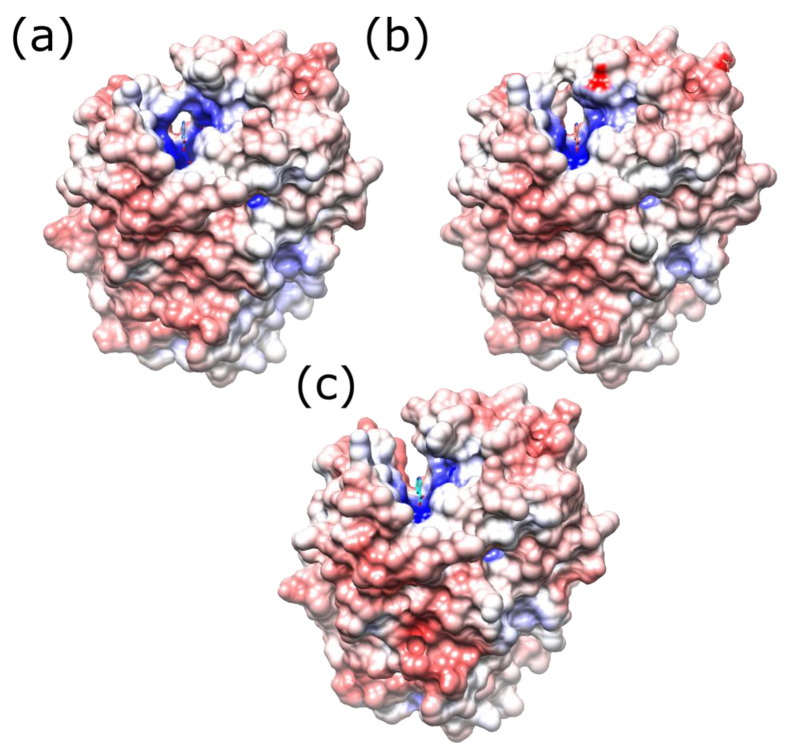
Electrostatic surface representation of wild-type and mutant NfsB, looking into the minor channel to the active site. (**a**) Wild-type protein; positively charged residues are coloured blue, negatively charged ones are coloured red, and the structure of the bound nicotinate is shown in blue with CPK colouring for the oxygen and nitrogen. (**b**) T41Q/N71S/F124T NfsB and (**c**) T41Q/N71S/F124T/M127V NfsB in the same orientation as the wild-type protein.

**Figure 8 ijms-24-05987-f008:**
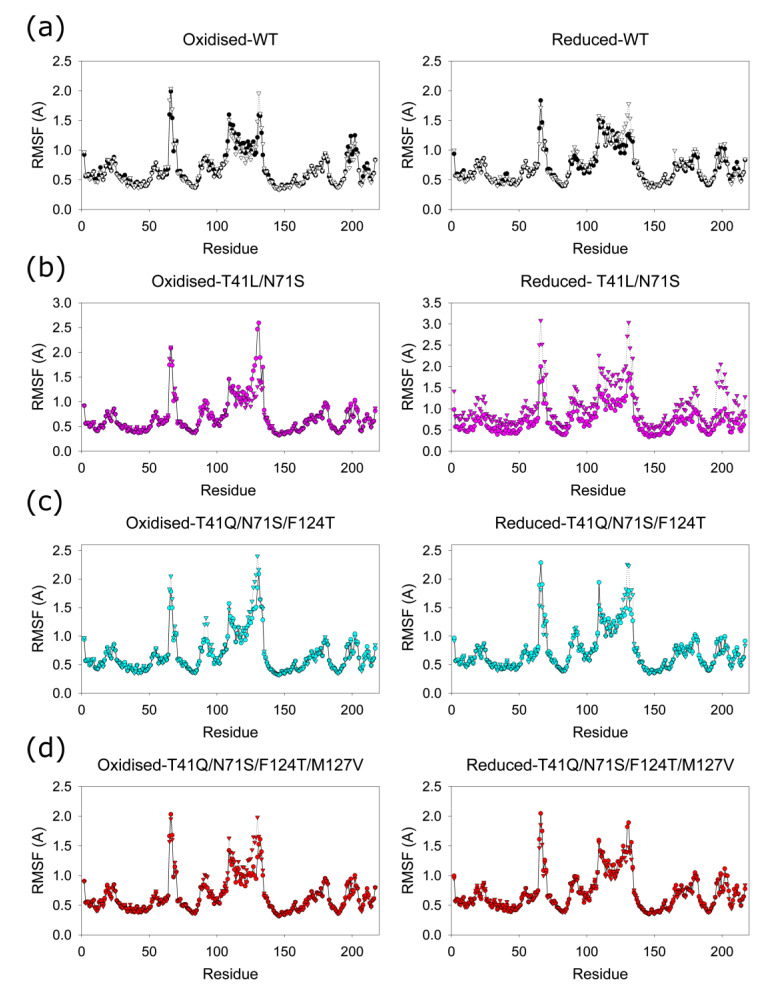
Root-mean-square fluctuations of Cα atoms in wild-type and mutant proteins. Each symbol is the average of the RMSFs in three runs. Circles are for subunit 1, and triangles are for subunit two. Left shows the enzymes with oxidized cofactor (FMN) in the active sites, and right is for enzymes with reduced cofactor (FMNH^−^) in the active sites. (**a**) Black and white symbols, wild-type enzyme, (**b**) magenta symbols, T41L/N71S enzyme, (**c**) cyan symbols, T41Q/N71S/F124T enzyme, and (**d**) red symbols, T41Q/N71S/F124T/M127V.

## Data Availability

The crystal structures have been deposited in the PDB as 8C5E- T41Q/N71S/F124T NfsB with nicotinate, 8C5F- T41Q/N71S/F124T NfsB, 8C5P-T41L/N71S NfsB, 8CCV- T41L/N71S NfsB with nicotinate, 8CJ0- T41Q/N71S/F124T/M127V NfsB with nicotinate. An additional structure of T41Q/N71S/F124T NfsB with citrate has also been deposited in the PDB as 8OG3. The molecular dynamics models are available in Model Archive (modelarchive.org./doi/10.5452) accession codes ma-uf6ua (wild type oxidized), ma-heo0p (wild-type reduced), ma-safhws (T41L/N71S oxidized), ma-tagoa (T41L/M71S reduced), ma-b37dk (T41Q/N71S/F124T oxidized, ma-eavjc (T41Q/N71S/F124T reduced), ma-tc4tm (T41Q/N71S/F124T/M127V oxidized), and ma-z0zpq (T41Q/N71S/F124T/M127V reduced).

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
