# Peer review of "Structure and Dynamics of Three Escherichia coli NfsB Nitro-Reductase Mutants Selected for Enhanced Activity with the Cancer Prodrug CB1954"

_ijms, 2023, doi:10.3390/ijms24065987_

Round 1

Reviewer 1 Report

E. coli NfsB is a flavoenzyme that reduces a variety of molecules, including the prodrug CB1954. Upon reduction by NfsB, the active form of CB1954 crosslinks DNA leading to cancer cell death. To improve the effectiveness of NfsB/CB1954 and therefore its applicability for clinical trials, different labs have been studying NfsB mutants with goal to enhance the binding potency/activity of the enzyme against CB1954. As part of this effort, the current manuscript describes the characterization of three NfsB mutants (T41L/N71S, T41Q/N71S/F124T, and T41Q/N71S/F124T/M127V) that exhibit improved functional features in contrast to WT protein and previously reported mutants.

Without questioning the accuracy of the experimental findings, I believe that the manuscript lacks novelty. Whereas several single, double, and triple mutants of NfsB have been previously analyzed by protein crystallography, the crystal structures of T41L/N71S and T41Q/N71S/F124T do not add a significant value to what is already known for the enzyme. The redox plots show some differences, in comparison to WT protein, but these are not drastic. Similar to crystallography and redox studies, the MD simulations demonstrate minor differences among the reported mutants and WT enzyme. For those reasons, the manuscript is premature and not suitable for publication to IJMS.  

Author Response

While the data do not show any dramatic effects with the mutations, this is a result in itself, and we believe that this paper gives new information and insight into the system. Together with our previous paper, (Jarrom et al. 2009) it is a comprehensive examination of 2 mutants – T41L/N71S and T41Q/N71S/F124T and shows why there is synergy between the mutation T41Q, which otherwise is a detrimental mutation and F124T, which is less beneficial than other mutations at F124, resulting in all further mutants we selected for greater activity for CB1945 having these two mutations at these positions. The combination of the kinetic, redox, structural and dynamic studies show how the mutations affect the activity of both prodrug and NADH. They are the first studies to analyse the effects of mutations on the cofactor as well as on the substrates. They show that the redox potential of the protein is only slightly affected by the N71S mutation. However, while the selected mutations enhance the kinetics of this enzyme with prodrug they reduce the rate with the cofactor. This will eventually be limiting for enhancement of activity for any enzyme with a substituted enzyme mechanism, where the cofactor and substrate share the same active site. There is little change in backbone structure or backbone dynamics of this enzyme on mutation, but the residues that show the most fluctuations are those lining the active site of the enzyme that control substrate and cofactor binding. These fluctuations, which will be larger for the side chains, may allow the broad substrate range of this enzyme. To our knowledge, this is the first study to comment on the dynamics of the protein and its role in substrate specificity. Finally, we have selected and characterised a quadruple mutant with even higher activity for CB1954 than our previous mutants. We show that this mutation has enlarged a small second channel to the active site and show how its kinetics leads to higher activity. These results have implications for other mutants of NfsB, some of which are discussed in the text, as well as for other enzymes with a substituted enzyme mechanism.

We have slightly changed the abstract to bring out these findings.  

Reviewer 2 Report

Article: “1Structure and dynamics of three E. coli NfsB nitroreductase 2mutants selected for enhanced activity with the cancer prodrug 3CB1954”.

1. The work is good, consistent and targeted.

2. There are several abbreviations in the manuscript without meanings; I suggest adding section for abbreviations meanings or explanation; e.g. NfsB. 

3. Line 108: species name ”Fisheri”  is better to  write in small letters.

4. Line 518: sometimes “nfsB” is written in italic other times in normal form. Check to write this in a uniform form.

Author Response

  1. 1. we thank the reviewer for this comment.
  2. 2. There didn’t seem to be a section for abbreviations in the template, we have added the section shown to the manuscript, placing it  below the keywords:

 Abbreviations used: CB1954, 5[aziridin-1-yl]-2,4,dinitrobenzamide; MD, molecular dynamics; NFZ, nitrofurazone, RMSD, root mean square deviation; RMSF, root mean square fluctuation.

  1. Corrected
  2. I don’t know exactly what line is referred to here as I do not have any line numbers. NfsB with capital N and latin letters is the enzyme, nfsB is the gene. I have tried to be consistent with this. With the lysogens sometimes the text is about the gene and sometimes about the expressed enzyme.

Reviewer 3 Report

Manuscript deals with the structural analysis of several mutants of E. coli NfsB with enhanced activity for the prodrug CB1954, that have been characterized previously by their activity in vitro and in vivo.

Authors have shown the most active mutant selected was T41Q/N71S/F124T. While the N71S is the only mutation at position 71 that enhances the activity of the protein for CB1954, the F124T mutation is not the most active mutation at position 124 and the T41Q mutation by itself reduces the activity of the protein for CB1954. The mutations together therefore must show synergistic effects, and the structure of the triple mutant shows interaction between Q41 and T124, explaining the synergy between these two mutations.

However, they have shown the mutant protein has lower redox potentials than wild type NfsB and that for the mutant the reduction of the enzyme by NADH, rather than the reaction with CB1954, is the slower step of the reaction; in contrast to the wild type enzyme, making difficult to understand the enhanced activity of the protein for CB1954.

They analyzed also the most active mutant in E. coli T41Q/N71S/F124T/M127V, in which the additional M127V mutation enlarges the small channel to the active site. The mutant has a higher kcat than previous mutants, while its Km for CB1954 is much lower than for wild type NfsB, but between that of the double and triple mutant.

But the big bottle neck is that the Michaelis constant of the enzyme for the prodrug is much higher than the maximum prodrug concentration in serum, How good is this improvement of the mutant?

MD simulations show that the mutations or reduction of the FMN cofactors of the protein has little effect on its dynamics, and they speculate that the enhancements must be largely due to small changes in the orientations of the side chains and surrounding residues.

They said that despite the lower dissociation constants of the mutants for CB1954 than that of the wild type, they were unable to obtain crystals of the proteins with bound CB1954, and they speculate that the prodrugs only bind to reduced enzyme or with NAD(P)+ in one of the two active sites. As the crystal structure of the protein with bound prodrug is critical to clarify the enhanced activity of the mutants, more work or at least some modelling with the interactions with the reduced enzymes are required.

Author Response

1. The overall activity of the enzyme will depend on the slower of the rates of two steps, the first with NADH and the second with the prodrug. The mutations reduce the maximum rate with NADH, but increase the maximum rate with CB1954. In vivo the concentration of NADH is relatively high compared to the achievable concentrations of CB1954, so, despite this step being much slower than for wild-type enzyme, at low CB1954 concentrations it will still be faster than the second step for all the proteins. However, the mutations have improved the rate of the second step, so, under these conditions, where the rate depends on the second step, the mutants have improved activity for CB1954.  It is not clear how much further improvement could occur as, if other mutations further reduce the rate of the NADH reaction, the first reaction will become rate limiting at lower CB1945 concentrations.

We have made slight alterations to this paragraph in the discussion to make this point clearer.

2. We have shown here that this quadruple mutant is much more active with CB1954 and makes E. coli about 50 times more sensitive to CB1954 than wild type, killing half the colonies at 4.5 µM CB1954, even though its Km for CB954 is about 100-fold higher (Figure 5, Supplementary Table S4, S5). Given that wild-type enzyme shows some efficacy in vivo, such an improvement would be substantial. However, as shown in our previous work with double mutants, (Jaberipour et al. 2010)  the effect in human cells may be different, possibly due different expression levels or to other competing substrates, and needs further assays which we cannot now do. 

We have added a comment about its relative efficacy to the discussion.

3. AJC intends to do modelling and MD studies of the mutants with CB1954 and/or NADH or NAD+ in different combinations in the active sites. This will be a large body of new work, similar to the previous work with wild type E. cloacae enzyme (Christofferson 2020), and is beyond the scope of the current paper.

Reviewer 4 Report

This manuscript describes the study of structure and dynamics of three E. coli NfsB nitroreductase mutants for enhanced activity with a cancer prodrug. I think it is a well-designed study. It is noted that with wild type NfsB, at about 210 mM CB1954, only 50% of the original numbers of colonies grew. It was found that the absolute values for the IC50s varied only slightly and the relative efficacies of the mutants remained constant on repeating the assay on different days. The discussion was thoughtful, but the manuscript lacked a conclusion section. I think the research would be attractive to readers of IJMS. Thus, I recommend publishing it in IJMS after this minor change.

1.      Add a section for conclusion: summarize and conclude the research.

Author Response

We have added a conclusion section which summarises and reiterates the key points of the discussion, as suggested.

Reviewer 5 Report

The manuscript describes X-ray structure analysis and molecular dynamics simulation of three mutated NfsB proteins. Although I feel a little bit disappointed that docking studies with CB1954 are omitted, I think it is an interesting story and well organized. Two minor changes are needed as follows.

1. On line 33, “b-bi” should be “bi-bi”.

2. The yellow dashes in Figure 1 are too thin.

Author Response

  1. corrected
  2. Figure 1 has been redrawn, with the H-bonds in black continuous, wider, lines so that they show better, and including 2 residues mentioned in the text, as well as the mutated residues.
  3. As indicated for reviewer 3- AJC intends to do modelling and MD studies of the mutants with CB1954 and NADH in different combinations. This will be a large body of new work, similar to the previous work with wild type E. cloacae enzyme (Christofferson 2020), which should yield more detailed and comprehensive results than simple docking models, and is beyond the scope of the current paper.

Round 2

Reviewer 1 Report

I have not any major comments, the manuscript can be published in the present form.

Reviewer 3 Report

The authors answer only partially to the requested changes.